# HDAC6 Regulates Radiosensitivity of Non-Small Cell Lung Cancer by Promoting Degradation of Chk1

**DOI:** 10.3390/cells9102237

**Published:** 2020-10-04

**Authors:** Niko Moses, Mu Zhang, Jheng-Yu Wu, Chen Hu, Shengyan Xiang, Xinran Geng, Yue Chen, Wenlong Bai, You-Wei Zhang, Gerold Bepler, Xiaohong Mary Zhang

**Affiliations:** 1Cancer Biology Graduate Program, Department of Oncology, Wayne State University School of Medicine, Karmanos Cancer Institute, Detroit, MI 48201, USA; nmoses@med.wayne.edu; 2Molecular Therapeutics Program, Department of Oncology, Karmanos Cancer Institute, Wayne State University School of Medicine, 4100 John R. Street Detroit, MI 48201, USA; zhangm@karmanos.org (M.Z.); Jheng-Yu.Wu@moffitt.org (J.-Y.W.); huc@karmanos.org (C.H.); beplerg@karmanos.org (G.B.); 3Department of Pathology & Cell Biology, Morsani College of Medicine, University of South Florida, 12901 Bruce B. Downs Blvd., Tampa, FL 33612, USA; Shengyan.Xiang@moffitt.org (S.X.); wbai@usf.edu (W.B.); 4Department of Pharmacology, Case Comprehensive Cancer Center, School of Medicine, Case Western Reserve University,2109 Adelbert Road, Wood Building W343A, Cleveland, OH 44106, USA; xxg204@case.edu (X.G.); yxz169@case.edu (Y.-W.Z.); 5Department of Biochemistry, Molecular Biology and Biophysics, The University of Minnesota at Twin Cities, Minneapolis, MN 55455, USA; yuechen@umn.edu

**Keywords:** histone deacetylase 6 (HDAC6), checkpoint kinase 1 (Chk1), ubiquitination, ubiquitin E3 ligase, ionizing radiation (IR), DNA damage response (DDR)

## Abstract

We have previously discovered that HDAC6 regulates the DNA damage response (DDR) via modulating the homeostasis of a DNA mismatch repair protein, MSH2, through HDAC6’s ubiquitin E3 ligase activity. Here, we have reported HDAC6’s second potential E3 ligase substrate, a critical cell cycle checkpoint protein, Chk1. We have found that HDAC6 and Chk1 directly interact, and that HDAC6 ubiquitinates Chk1 in vivo and in vitro. Specifically, HDAC6 interacts with Chk1 via the DAC1 domain, which contains its ubiquitin E3 ligase activity. During the cell cycle, Chk1 protein levels fluctuate, peaking at the G2 phase, subsequently resolving via the ubiquitin-proteasome pathway, and thereby allowing cells to progress to the M phase. However, in HDAC6 knockdown non-small cell lung cancer (NSCLC) cells, Chk1 is constitutively active and fails to resolve post-ionizing radiation (IR), and this enhanced Chk1 activity leads to preferential G2 arrest in HDAC6 knockdown cells accompanied by a reduction in colony formation capacity and viability. Depletion or pharmacological inhibition of Chk1 in HDAC6 knockdown cells reverses this radiosensitive phenotype, suggesting that the radiosensitivity of HDAC6 knockdown cells is dependent on increased Chk1 kinase activity. Overall, our results highlight a novel mechanism of Chk1 regulation at the post-translational level, and a possible strategy for sensitizing NSCLC to radiation via inhibiting HDAC6’s E3 ligase activity.

## 1. Introduction

Histone Deacetylase 6 (HDAC6) is a unique class IIb HDAC [1,2], whose dissimilarities from conventional HDACs lie in its two tandem deacetylase domains (DAC1 and DAC2), primarily cytoplasmic localization, and the ubiquitin-binding property [3]. HDAC6’s canonical deacetylation targets include cortactin [4], α-tubulin [5], and HSP90 [6]. HDAC6 was first identified as a deacetylase of α-tubulin in a study that described its regulation of cell migration and motility [5], opening up the floodgates to a wave of subsequent studies detailing the oncogenic role that HDAC6 plays in a variety of tumor types, including breast cancer [7], ovarian cancer [8], AML [9], and glioblastoma [10]. Our published tissue microarray data also demonstrates that HDAC6 is upregulated across all three subtypes of non-small cell lung cancer (NSCLC) [11], the second most commonly diagnosed cancer across both sexes that is responsible for the greatest number of cancer-related deaths annually. Further clinical innovation is needed to improve outcomes for NSCLC patients.

HDAC inhibition as an anticancer therapeutic strategy has been gaining traction over the last two decades, and has seen major success in the FDA approval of pan-HDAC inhibitor Vorinostat for treatment of cutaneous T cell lymphoma (CTCL) [12]. It has been widely observed that pan-HDAC inhibitors can promote growth arrest, differentiation, and apoptosis in tumor cells with minimal off-target toxicity to the surrounding normal tissue [13]. These studies have promoted investigation into the particular HDACs responsible for the differential response between transformed tissue and normal tissue, as inhibition of these HDAC isotypes will enhance tumor responsiveness to intervention, while further mitigating off-target effects of the treatment [14]. In particular, HDAC6 knockout mice develop normally, which makes HDAC6 an ideal target in cancer treatment [15]. HDAC6-specific inhibition has been tested both pre-clinically and clinically, and a trend has emerged for combining HDAC6 inhibition with DNA damaging agents [16,17,18,19,20]. Currently there are two HDAC6-specific inhibitors in Phase I/II clinical trials, ACY-1215 and ACY-241, but much remains to be discovered concerning how HDAC6 interacts with DNA damage response (DDR) proteins and how HDAC6 inhibition impacts tumor DDR activity. We previously found that knockdown of HDAC6 in NSCLC cell lines sensitized these cells to cisplatin treatment [21]. Our current study is the product of linking and expanding our finding that HDAC6 knockdown NSCLC cells exhibit the enhanced cisplatin sensitivity [21] and our 2014 study, which describes E3 ubiquitin ligase activity contained in HDAC6’s DAC1 domain [22]. We identified a DNA mismatch repair protein, MSH2, as a target of HDAC6’s novel E3 ligase activity and deacetylase activity. We also found that loss of HDAC6 subsequently increases the level of MSH2, leading cells to become sensitive to 6-thioguanine (6-TG) treatment. However, we suspect that HDAC6 may play a more overarching role in the regulation of the DDR via ubiquitination of currently unidentified targets.

Chk1 is a Ser/Thr kinase activated by ATR in response to a variety of DNA aberrations, including UV-light induced damage, DNA interstrand crosslinks, and the resected ends of a DNA double-strand break [23]. Specifically, ATR phosphorylates Chk1 on canonical Serine (Ser) residues 317 and 345, which activate Chk1’s kinase activity and renders it susceptible to degradation via the ubiquitin-proteasome pathway [24,25,26,27]. Active Chk1 stimulates the S and G2 checkpoints of the cell cycle through inhibitory phosphorylations to phosphatases CDC25A [28,29] and CDC25C [30], respectively. The fluctuation of total Chk1 protein levels peaking at S/G2 and decreasing as the cell prepares to re-enter G1 is well documented [31], and multiple E3 ligases have been identified that contribute to Chk1 degradation [32,33,34,35]. It is fairly common for central DDR proteins to act under the tight control of multiple E3 ligases [36], p53 being the most notable example [37], which suggests that the complete repertoire of Chk1-targeting E3 ubiquitin ligases has yet to be uncovered. In this report, we propose HDAC6 as a candidate Chk1 E3 ubiquitin ligase, as we have confirmed that HDAC6 and Chk1 interact and that HDAC6 ubiqutinates Chk1 in NSCLC cells and in vitro. We have also found that in addition to DNA-damaging chemotherapeutics, genetic ablation of HDAC6 can increase the efficacy of ionizing radiation (IR). Further analysis of irradiated NSCLC cells revealed that in HDAC6-knockdown cells the stable Chk1 is active, as indicated by phosphorylation on Ser317 and Ser345. While elevated Chk1 activity has previously been reported to contribute to tumoral genomic stability [38,39], constitutive Chk1 expression and activation resulting from HDAC6 knockdown appears to serve as a detriment to the ability of irradiated NSCLC tumor cells to withstand ionizing radiation.

## 2. Materials and Methods

### 2.1. Antibodies, Chemicals, and Reagents

Anti-Flag M2 agarose beads (A2220) and anti-HA agarose beads (A2095) were purchased from Sigma. Anti-Chk1 (sc-56288) and anti-p53 (sc-55476) antibodies were purchased from Santa Cruz. Anti-PARP-1 (9532), anti-pChk1S345 (2341), anti-pChk1S317 (2344), anti-p-p53 (9284), anti-γ-H2AX (9718), anti-pATR-S428 (2853), anti-pATM-S1981 (13050) and anti-pCdc25C (9528S) antibodies were purchased from Cell Signaling Technologies. Cycloheximide (C4859), cisplatin (479306), and imidazole (12399) were purchased from Sigma. Ni-NTA resin (635659) was purchased from Clontech.

### 2.2. Establishment of HDAC6 Knockout Cell Lines

HDAC6 knockout (KO) cells were created using CRISPR/Cas9 (Clustered regularly interspaced short palindromic repeats) method. Briefly, the guide RNA targeting HDAC6 exon 5 (5′-GAAAGGACACGCAGCGATCT-3′) was selected and constructed into LentiCRISPRv2 vector (Addgene plasmid 52961). The HDAC6-KO vector can also express the codon-optimized Cas9 protein as well as puromycin resistance gene. The 293T, NCI-H1975, and NCI-H157 cells infected with the HDAC6-KO virus were selected for stable clones using puromycin at 1 μg/mL. The HDAC6-KO clones were screened by anti-HDAC6 (H-300, Santa Cruz, CA, USA) Western blot analysis.

### 2.3. Cell Culture

The NSCLC cell lines A549, NCI-H460, NCI-H157, NCI-H1975, and NCI-H1299 cells were obtained from the American Type Culture Collection (ATCC, Gaithersburg, Maryland, USA). All cell lines were grown in either Dulbecco’s Modified Eagle’s Medium (DMEM) or Roswell Park Memorial Institute 1640 Medium (RPMI), both with 10% fetal bovine serum, penicillin (100 U/mL), and streptomycin (100 U/mL). All cells were incubated at 37 °C with 5% CO_2_. 293T and 293T HDAC6 KO cells were grown in DMEM. NCI-H1299, A549, NCI-H460, NCI-H157, NCI-H1975, and NCI-H1299 cells were grown in RPMI. A549 HDAC6 knockdown (KD) and control stable cell lines were kindly provided by Dr. Tso-Pang Yao from Duke University. NCI-H460 scramble and HDAC6-KD stable cells were generated as previously described [21]. For simplicity, NCI-H1299, NCI-H460, NCI-H157, and NCI-H1975 are referred to as H1299, H460, H157, and H1975 hereafter.

### 2.4. Generation of H1299 and A549 Inducible HDAC6 Knockdown Cells

The shRNA vectors against HDAC6 were ordered from Dharmacon (cat. RHS4696), transfected into H1299 and A549 cells, and selected with 1μg/mL puromycin for 1 month. Independent clones were then chosen by induction with 0.1 μg/mL doxycycline (D9891, Sigma, St. Louis, MO, USA) for 3 days, and then subjected to Western Blotting for HDAC6. All cells were cultured with tetracycline-free medium.

### 2.5. Generation of A549, H157, and H1975 HDAC6 Knockout Cells

HDAC6 knockout A549, H157, and H1975 cells were created using the CRISPR (Clustered regularly interspaced short panlindromic repeats) method. Briefly, the guide RNA targeting HDAC6 exon 5 (5′-GAAAGGACACGCAGCGATCT-3′) was selected and constructed into pLENTICRISPR-V2 vector (Addgene plasmid 52961). This HDAC6-KO vector can also express the codon-optimized Cas9 protein as well as puromycin resistance gene. After transfecting this KO vector into 293T cells together with pSPAX2 and PMD2.G packaging vectors at 2:1:1 ratio, the lentivirus medium was harvested 48 h after transfection. The KO lentiviruses were transduced into A549, H157, and H1975 with polybrene at 8 μg/mL and select the stable clones or pools with puromycin (the puromycin concentration should be determined based on the killing curve). The HDAC6-Knockout clones were screened by anti-HDAC6 (H-300) Western blot analysis and furthered confirmed by sequencing HDAC6 exon 5 region. A pair of primers were used to amplify exon5 region of HDAC6 (SR-exon5-F, GGGTTTTAACCAGTGTCTAGCTG; SR-exon5-R, TGGCTGCAGAGACCATGACATTG). The resulting PCR products were cloned into PCR-blunt vector (Invitrogen, Waltham, Massachusetts, USA) for sequencing both alleles in human genome. The confirmed HDAC6-KO clones were used for further experiments.

### 2.6. Irradiation

Radiation treatment was performed with an X-ray generating Pantak HF 320 instrument (Shimadzu Corporation, Kyoto, Japan) (settings at 320 kV, 10 mA, @ ~0.9 Gy per minute).

### 2.7. Trypan Blue Exclusion

Upon completion of a radiation time course, cells were trypsinized, suspended in PBS, and the density of the solution was quantified via hemacytometer. A 0.4% solution of trypan blue in PBS was prepared, and 0.1 mL of this trypan blue solution was added to 0.1 mL of cells in PBS. The trypan blue and cell mixture was incubated for 15 min at room temperature (RT), then loaded into a hemacytometer and manually counted under a microscope. The number of blue-stained cells and the number of total cells were recorded, and viability assessed as: Fraction of viable cells % = (1.00 − (number of blue cells ÷ number of total cells)) × 100.

### 2.8. Colony Formation Assay

For Figure 1D,E, A549 inducible (A549i) cells were treated with doxycycline (0.1 μg/mL) for 2 weeks to deplete HDAC6, and subsequently the cells were seeded in triplicate (300 cells/mL) into 6-well plates. Cells were then incubated overnight at 37 °C to allow for adherence to the dishes. The cells were then treated with the indicated doses of radiation and incubated for 13 days. Upon completion of the time course, cells were directly fixed and stained with crystal violet (0.05% *w*/*v*, 1% formaldehyde, 1% methanol in PBS) for 20 min. Colonies on each plate were scanned and counted using the Cell Counter feature of the Image J software. For Figure 6B,C, A549 HDAC6 stable knockdown (HDAC6KD) and Chk1-Tripz (HDAC6KD+Tripz) cells were seeded in triplicate (150 cells/mL) into 6-well plates, and otherwise treated and analyzed as described for the A549 inducible HDAC6 knockdown cells.

### 2.9. Constructs and Transfection

The GST-tagged HDAC6 deletion mutant constructs were synthesized as previously described [22]. The Myc-Chk1, Myc-Chk1(1–264) and Myc-Chk1 (265–476) plasmids were as described [40]. Flag-Chk1 was purchased from Addgene (22894). The plasmids were transiently or stably transfected into cells using Lipofectamine 2000 (Invitrogen).

### 2.10. GST Pull-Down Assay

BL21 cells harboring the GST or various GST recombinant HDAC6 plasmids were grown to log phase and induced with Isopropyl β-D-1-thiogalactopyranoside (IPTG) for 4 h. After sonication in STE buffer (10 mM Tris-HCL (pH 8.0), 150 mM EDTA, and 5 mM dithiothreitol (DTT)) containing 1% sarcosyl (*w*/*v*, final concentration), solubilized proteins were recovered by centrifugation and incubated with glutathione-agarose beads in the presence of 3% Triton X-100 (final concentration) for 30 min at 4 °C.

### 2.11. Generation of Chk1 Inducible Knockdown Cells

TRIPZ inducible lentiviral shRNA against Chk1 was purchased from Dharmacon (RHS4696 glycerol stock). The TRIPZ plasmid was transfected into 293T cells along with lentiviral packaging and envelope (2nd generation) plasmids (Addgene packaging 11263, Addgene envelope 17576). 48 h post-transfection, the DMEM media was collected from the 293T cells, and 200 μL of this media was added to A549 HDAC6 stable knockdown cells at ~60% confluence in a 6 cm dish. These A549 cells were expanded and split into a 10 cm dish, at which point doxycycline (0.1 μg/mL) was added. Cells were treated with doxycycline for 2 weeks to induce the expression of the TRIPZ vector, which will co-express shRNA against Chk1 and fluorescent marker TurboRFP. RFP-positive cells were sorted using a Sony SY2300 (Sterile/4-Way) cell sorter incorporated into a Baker SterilGARD II Biological Safety Cabinet, allowing for sterile BSL Class II sorting. This sorting produced the Chk1Tripz pool. Further experiments were conducted after establishment of stable clones with high expression of RFP.

### 2.12. RT-PCR

Reverse transcriptase-polymerase chain reaction (RT-PCR) assays were performed to measure the expression of mRNA. Cells were washed at least twice with PBS and immediately lysed in Trizol^®^ (AMBION, Catalog number: 15596026). For Mouse tissue samples: Add 1 mL of TRIzol™ Reagent per 50–100 mg of tissue to the sample and homogenize using a homogenizer. Total RNA was then isolated to follow the Trizol method (Invitrogen). Subsequently, 1 μg of RNA was reverse-transcribed using the WarmStart^®^ RTx reverse transcriptase (New England BioLabs, M0380L, Ipswich, MA, USA) and random primer mix (New England BioLabs, S1330S) according to the typical cDNA synthesis protocol. PCR reactions were performed with Taq 2X Master Mix (New England BioLabs, M0270L). The thermocycler conditions for Mouse Chk1 and GAPDH PCR product were as follows: 95 °C 30 sec for 1 cycle; 95 °C 30 sec, 55 °C 60 sec, and 68 °C 1 min for 40 cycles; final extension 68 °C 5 min for 1 cycle. The same thermocycler conditions were used for Human Chk1 and GAPDH. The following PCR primers were used for RT-PCR: Human Chk1-forward: 5′-ATGCTCGCTGGA GAATTGC-3′; Human Chk1-reverse: 5′-ATAAGGAAAGACCTGTGCGG-3′; Human GAPDH-forward: 5′-GGAGCGAGATCCCTCCAAAAT-3′; Human GAPDH-reverse: 5′-GGC TGTTGTCATACTTCTCATGG-3.′; Mouse Chk1-forward: 5′-CTTTGGGAGAAG GTG CCTAT-3′; Mouse Chk1-reverse: 5′-ATGCCGAAATACCGTTGC-3′; Mouse GAPDH-forward: 5′-GTTGTCTCCTGCGACTTCA-3′; Mouse GAPDH-reverse: 5′-GGTGGTCCAGGGTTTCTTA-3.′ The PCR products were then loaded onto the agarose gel with ethidium bromide. After gel electrophoresis, the PCR products were visualized under the UV light.

### 2.13. In Vivo Ubiquitination Assay

Myc-Chk1 (4 μg) was cotransfected with 2 μg of Flag-HDAC6 and His-Ub as indicated into 293T cells. 36 h post-transfection, cells were harvested. Cell lysates were lysed in 6 M guanidine buffer (50 mM Na_2_HPO4, 50 mM NaH_2_PO4, 10 mM Tris-HCl (pH 8.0), 10 mM β-mercaptoethanol, 5 mM imidazole, 0.2% Triton X100) overnight with Ni-NTA beads. The beads were washed with 8 M urea buffer (otherwise same components of the guanidine buffer; 50 mM Na_2_HPO4, 50 mM NaH_2_PO4, 10 mM Tris-HCl (pH 8.0), 10 mM β-mercaptoethanol, 5 mM imidazole, 0.2% Triton X100) 1 time, then washed with 8 M urea buffer (pH 8.0) 3 times. The beads were eluted with elution buffer (0.5 M imidazole, 0.125 M dithiothreitol (DTT), 1XSDS loading buffer) for 30 min at room temperature. The beads slurry were denatured at 95 °C for 5 min before loading.

### 2.14. In Vitro Ubiquitination Assay

The in vitro Ub assays were carried out in the presence of E1, E2, Ub, His-Chk1, Flag-HDAC6 in the absence or presence of ATP. The reactions were incubated at 37 °C for 2 h, denatured at 95 °C for 5 min, then added protein loading buffer. The reactions were loaded on SDS-PAGE followed by Western blotting analysis with the anti-Chk1 antibody. The detailed protocol is described in Zhang et al. [22]

### 2.15. Animals

All animal-related procedures and protocols were performed under the Institutional Animal Care and Use Committee (IACUC) at Wayne State University (WSU). HDAC6 wild-type and HDAC6 knockout mice in a C57BL/6 background were originally obtained from Dr. Patrick Matthias’ lab at the Friedrich Miescher Institute for Biomedical Research (FMI) in Basel, Switzerland. All mice were under 24 hrs/day, 7days/week veterinary care and provided food and water *ad libitum.* Primary and secondary methods of euthanasia (CO2 and cervical dislocation, respectively) were followed by tissue harvest.

## 3. Results

### 3.1. HDAC6 Depletion Sensitizes Several NSCLC Cell Lines to IR

The first question to be answered is whether HDAC6 knockdown sensitizes NSCLC cells to IR. We previously found that HDAC6 knockdown preferentially sensitizes cells to cisplatin treatment; this sensitization was presumed to be mechanism-specific, as parallel treatment with paclitaxel did not further sensitize HDAC6 knockdown cells [21]. While the interstrand DNA crosslinks induced by cisplatin differs from the single- and double-strand DNA breaks IR generates, we suspect that the efficacy of treatment in HDAC6 knockdown cells relies on direct DNA damage. Paclitaxel is a cytoskeletal drug that inhibits spindle formation, and the sensitivity of control cells and HDAC6 knockdown cells to paclitaxel treatment is identical. HDAC6 knockdown cells may be more sensitive to cisplatin due to HDAC6’s regulatory interaction with MMR proteins MSH2, MSH6, and MLH1, and these cells may also be more sensitive to IR due to HDAC6’s interaction with DNA double-strand break sensors MRE11 and RAD50 [22].

We assessed the viability of A549 and H460 stable HDAC6 knockdown cells and H1299 HDAC6 inducible knockdown cells via trypan blue exclusion. Our A549 stable HDAC6 knockdown cells were treated with the indicated doses of radiation and incubated for 120 h, at which point they were harvested and stained with trypan blue exclusion dye (Figure 1A). We found a significant and dose-dependent reduction in viability in the HDAC6 knockdown cells, and confirmed this reduction in H460 HDAC6 stable knockdown cells and H1299 HDAC6 inducible knockdown cells 120 h post-10 Gy radiation (Figure 1B,C). These data indicate that regardless of whether HDAC6 knockdown is inducible or stable, its loss contributes to exacerbated reductions in viability of NSCLC cells upon IR.

We next sought to determine the survival of NSCLC cells post-radiation via colony formation assay, a gold standard for assessing radiation efficacy. Initial assessment of the A549 and H460 HDAC6 stable knockdown cells revealed a differential plating efficiency between the controls and the HDAC6 knockdowns, making a case for HDAC6 downregulation as a single treatment modality but inconclusive when assessing radiosensitization (data not shown). To subvert this shortcoming of stable knockdown cell lines, A549 inducible HDAC6 knockdown cells were used for the same assay. These cells successfully corrected for the plating efficiency error, and demonstrated a preferential reduction of colonies in the HDAC6-knockdown cells at 1, 2, 3, 4, and 5 Gy (Figure 1D,E), suggesting that depletion of HDAC6 causes a reduction of a short-term as well as a long-term survival post-IR.

### 3.2. HDAC6 Knockdown A549 Cells Exhibit Prolonged G2 Arrest upon IR

To further assess survival of our HDAC6 knockdown cells post-radiation, we used propidium iodide (PI) staining to determine the fraction of cells presenting in the sub-G1 stage of the cell cycle, as a distinct sub-G1 peak indicates a population of dead cells [41]. Consistent with the data shown in Figure 1, we observed significantly larger sub-G1 population in our A549 HDAC6 knockdown cells than their control counterparts 72 h post-IR (Figure 2A,B). In terms of the time point, we selected 72 h post IR for our PI analysis, as we were informed by the MTT assays (data not shown). We wanted to capture cell cycle arrest beyond 24 h, when early cell death is occurring, but avoid the induction of late cell death at 120 h as indicated by our trypan blue exclusion assay (Figure 1A) (data not shown).

Interestingly, we also observed an increase in G2 arrest in the HDAC6 knockdown cells compared to the controls (Figure 2A,C). We confirmed this preferential arrest within the G2 phase with immunofluorescence staining for cyclin A, a critical cyclin for late-S and G2/M-phase cells [42], and revealed a preferential accumulation of cyclin A in HDAC6 knockdown cells at later time points post-IR (Figure 2D,E). The accumulation of G2-phase cells staining positive for Cyclin A led us to ponder whether Chk1, the gatekeeper of S and G2 phases of the cell cycle [28,43], was involved in the differential response of control and HDAC6 knockdown cells to radiation.

### 3.3. Chk1 Is Constitutively Active in HDAC6 Knockdown Cells Post-DNA Damage

In our 2012 article detailing increased sensitivity of HDAC6 knockdown cells to cisplatin, we found an increased activation of Chk1 exhibited by the HDAC6 knockdowns along with a reduction in cell viability post-cisplatin treatment [21]. To investigate whether a similar mechanism was occurring in the irradiated HDAC6 knockdown cells, we treated A549 stable HDAC6 knockdown cells with 10 Gy of IR and assessed our previous panel of intracellular markers of the DDR: pATR, pATM, p53, Chk1 and γ-H2X (Figure 3). We saw no difference in the induction of pATM (Figure 3A,D), which classically detects DNA DSBs [44], but found a preferential and persistent induction of pATR in our HDAC6 knockdown cells when compared to the controls (Figure 3A,C). We also observed preferential phosphorylation of p53 on Serine 15, a marker of cell death [45], in the HDAC6 knockdowns (Figure 3A,E). Next, we assessed protein levels of total Chk1, active Chk1, and γ-H2AX as a surrogate for double strand breaks. We were able to confirm Chk1’s enhanced stability in the knockdowns (Figure 3A,I), and found that both pChk1 S317 and pChk1 S345 were readily detectable throughout the time course (1, 6, 24, 36, 48 hr), consistent with increased activation of ATR (Figure 3B,G, and H). This persistence of active Chk1 occurred in parallel to the inability of these cells to resolve DNA DSBs, as indicated by γ-H2AX persistence (Figure 3A,J).

Our IR-specific data begs the question of whether this enhanced Chk1 response is radiation-specific, or can be observed upon treatment with a broader spectrum of DNA-damaging agents. We tested a radiomimetic drug etoposide (ETO) in our A549 stable knockdown cells using a 20 μM dose, and found the same Chk1 activation and γ-H2AX persistence phenotype that occurred in our radiation time course study (Figure 4). To test an agent whose primary mechanism of action is not double strand break formation and to validate our earlier work [21], we conducted a cisplatin time course in the A549 stable knockdown cells, and detected the Chk1 activation and γ-H2AX persistence phenotype (Figure 5). The consistency of this response between DNA-damaging agents highlights a potential mechanism of cell death in the absence of HDAC6, which we hypothesize is dependent on the increased Chk1 protein which leads to increased Chk1’s kinase activity. Please note, in terms of the time points in Figure 3, Figure 4 and Figure 5, we selected two time courses: (1) 0, 1, 6, 48, and 72 h; (2) 0, 1, 6, 24, 36, 48 h for IR treatment; one time course: 0, 1, 6, 24, 36, 48 h for etoposide treatment; and one time course: 0, 24, 48, and 72 h for cisplatin treatment. We wanted to capture the activation of cell cycle arrest proteins beyond 24 h, when early cell death is occurring, but prior to the induction of late cell death at 120 h as indicated by our trypan blue exclusion assay (Figure 1A).

### 3.4. Radiosensitivity of HDAC6 Knockdown Cells Is Dependent on Increased Chk1 Activity

We have observed that HDAC6 knockdown can both promote accumulation of Chk1 (Figure 3, Figure 4 and Figure 5) and sensitize cell lines to radiation (Figure 1 and Figure 2). We hypothesize that these two phenomena are linked via a causal relationship, and have established this relationship using both genetic ablation and pharmacological inhibition of Chk1. To genetically downregulate Chk1, we generated an A549 HDAC6 stable and Chk1 inducible knockdown cell line by stably transfecting A549 HDAC6 knockdown cells with a TRIPZ inducible shRNA plasmid targeted against Chk1, referred to hereafter as HDAC6 KD + Tripz. In addition to directly confirming Chk1 and HDAC6 double knockdown in A549 cells by anti-Chk1 and anti-HDAC6 Western blotting analyses, we have shown that knockdown of HDAC6 alone increased the level of acetylated tubulin, Chk1, and pCDC25C (Figure 6A). Since tubulin is an HDAC6’s substrate [5], increased acetylated tubulin level suggests the decreased HDAC6 activity in HDAC6 knockdown cells. Because CDC25C is a Chk1 substrate [30], our results indicate that the depletion of HDAC6 mediated an increase in Chk1, which leads to increased CDC25C activation (presumably by Chk1). Furthermore, knockdown of Chk1 in HDAC6 depleted cells decreased both Chk1 and pCDC25C, suggesting that the deceased Chk1 level mirrors the decreased Chk1 activity (Figure 6A). Therefore, the above experiments have successfully validated the HDAC6 knockdown (HDAC6KD) and HDAC6 and Chk1 double knockdown A549 cells (HDAC6KD + Tripz cells), which would be used in the following assays. We then assessed long-term survival of HDAC6 + Tripz cells compared to HDAC6KD cell lines post-IR via colony formation assay. Using an escalating dose of IR (1, 2, 3, 4 and 5 Gy), we found that genetic ablation of Chk1 had a protective effect on the ability of these cells to survive post-radiation (Figure 6B,C). To test whether Chk1 ablation influenced cell cycle distribution in our HDAC6 KD + Tripz cells, we used immunofluorescence to assess cyclin A protein levels post-IR in both our single and double-knockdown cell lines. As expected, we found that fewer HDAC6 KD + Tripz cells were cyclin A positive than the single HDAC6 KD cells at every time point tested (i.e., 0, 24, 48, and 72 h) (Figure 6D,E), suggesting a decrease in G2 arrest. We were unable to assess the full cell cycle distribution of our HDAC6 KD + Tripz cells using PI staining, as these cells also express a Turbo RFP reporter that would interfere with the PI signal.

To test the impact of pharmacological inhibition of Chk1 on the radiosensitivity of A549 cells, we utilized CHIR-124, a novel, potent, and specific Chk1 inhibitor with 2000-fold less activity against Chk2 and 500–5000-fold less activity against CDK2/4 and Cdc2 [46]. We chose this inhibitor in lieu of clinically relevant inhibitor UCN-01 [47] due to the need for Chk1-specific inhibition over a kinetic profile suitable for use in humans. We pre-treated A549 HDAC6 knockdown cells with either CHIR-124 or DMSO (vehicle) for 24 h prior to treatment with 10 Gy radiation. PARP-1 cleavage was detecTable 24 h after radiation alone in the HDAC6 knockdown cells, as expected given our previous results demonstrating a preferential sensitivity of HDAC6 knockdown cells to radiation. However, pre-treatment with CHIR-124 protected these cells from radiation-induced PARP-1 cleavage (Figure 6F). Taken together, this data indicates that the preferential sensitivity of HDAC6 knockdown cells to IR-induced damage and subsequent death (Figure 1) could be due to the increased levels of active Chk1 present in these cells (Figure 3, Figure 4 and Figure 5). We hypothesize that the reason for the increased levels of total and active Chk1 is due to the elimination of a Chk1-targeting ubiquitin E3 ligase, HDAC6.

### 3.5. HDAC6 Influences Chk1 Protein Stability and Ubiquitinates Chk1

HDAC6 is unique amongst the HDAC family members in that it possesses both deacetylase activity and E3 ubiquitin ligase activity. While a myriad of substrates have been identified for HDAC6’s deacetylase activity, to date MSH2 is the only reported target of HDAC6’s E3 ubiquitin ligase activity [22]. We suspected that there might be additional targets of this activity, specifically proteins involved in the DDR, such as Chk1.

Our group has previously generated A549, H157, and H1975 HDAC6 knockout cells using the CRISPR/Cas9 system as described in the Methods. We began by probing these cell lines for HDAC6, acetylated α-tubulin (a downstream target of HDAC6 deacetylase activity), Chk1, and GAPDH. Across all three cell lines, successful HDAC6 knockout corresponded with an increase in baseline Chk1 protein (Figure 7A, lanes 1–6). Inducible HDAC6 knockdown cell lines H1299 and A549 were pre-treated with doxycycline for two weeks, and these cells exhibited complete (H1299) and partial (A549) HDAC6 knockdown accompanied by an increase in Chk1 protein levels in both cell lines (Figure 7A, lanes 7–10). Mouse embryonic fibroblasts (MEFs) obtained from C57Bl/6 wild-type and HDAC6 knockout mice were also tested, and we observed the same increase in Chk1 protein in the HDAC6 knockout MEFs compared to the controls (Figure 7A, lanes 11–12). Finally, we harvested a variety of organ samples from our C57Bl/6 wild-type and HDAC6 knockout mice, and across every tissue type we tested an absence of HDAC6 protein corresponded with a higher basal level of Chk1 than observed in HDAC6-competent mice (Figure 7A, lanes 13–24). The expression of Chk1 in the above paired cell lines and mouse tissues has been quantified and shown in Figure 7B.

To determine whether Chk1 expression is being regulated at the protein level or the mRNA level, RT-PCR for Chk1 was performed in our A549 stable knockdown cells as well as wild-type and HDAC6 KO murine lung tissue. Knockdown or knockout of HDAC6 did not influence Chk1 mRNA levels (Figure 7C), so we proceeded by examining the regulation of Chk1 expression at the protein level.

To determine whether the increase of Chk1 in HDAC6-knockdown cells is due to increased protein stability, we treated A549 stable knockdown cells with 10 μg/mL cycloheximide, a protein synthesis inhibitor, over an 18 h time course. Previous studies have reported a Chk1 half-life of ~4.6 h in parental cancer cells [33], so this time course should model typical Chk1 resolution in the control cells. Indeed, Chk1 degraded as expected in the control cells and was virtually undetecTable 18 h post-cycloheximide treatment. In contrast, Chk1 was exquisitely stable in the HDAC6 knockdown cells (Figure 7D). This protocol was repeated three times, and Chk1 resolution was exponentially modeled in each cell type (data not shown). Chk1 half-life in the control cells is ~5.4 h, while Chk1 half-life in HDAC6 knockdowns is ~40 h, indicative of Chk1’s significant stability in the absence of HDAC6. Beginning to explore whether HDAC6 could decrease the level of Chk1, we overexpressed HA-tagged HDAC6 in the 293T HDAC6 knockout cells described in Wu et al. [48]. Our untransfected cells maintained high acetyl-tubulin levels, and we used their expression of Chk1 as a control. When HDAC6 was overexpressed for 72 and 120 h, Chk1 protein levels were reduced by half and three-quarters, respectively, as compared to the control (Figure 7E), suggesting that HDAC6 is able to down-regulate protein levels of Chk1.

Our group previously uncovered ubiquitin E3 ligase activity within the DAC1 domain of HDAC6, which regulates MSH2 stability via ubiquitination both in vitro and in vivo [22]. The observed association between loss of HDAC6 and increased Chk1 stability may indicate a role for HDAC6 in the ubiquitination and subsequent degradation of Chk1. To test HDAC6’s intrinsic E3 ligase activity, we performed the in vitro Ub assays in the absence or presence of ATP, an essential component of this assay. As shown in Figure 8A, HDAC6 purified from 293T cells was able to efficiently ubiquitinate His-Chk1. The quality of purified Flag-HDAC6 and His-Chk1 proteins was shown in Figure 8B,C by coomassie blue staining. To test whether HDAC6 promotes Chk1 ubiquitination in cells, we performed a Ub assay under denatured conditions. As shown in Figure 8D, overexpression of HDAC6 in 293T cells greatly increased the level of ubiquitinated Chk1, suggesting that HDAC6 is able to ubiquintinate Chk1 in cell lines. We proceeded by assessing the impact of lysine residue K436, located just outside Chk1’s degron-like motif (368–421), on the ability of the total protein to be ubiquitinated. It has previously been reported that mutating this residue to arginine (K436R) enhances Chk1 stability [32], so we wanted to confirm that residue is involved in the ubiquitination of Chk1. To test the relevance of this residue on the stability of Chk1 in NSCLC cells, we transfected H460 cells with either Myc-Chk1 or Myc-K436R mutant Chk1, and treated these cells with CHX for 0, 3, 6, and 24 h. We found that the mutant was more stable than wild-type Chk1 (Appendix A). Finally, we performed a Ub assay under denatured conditions using either wild-type Myc-Chk1 or Myc-K436R mutant Chk1 as a substrate, and found that mutation of this residue reduced ubiquitination of Chk1 (Appendix A).

### 3.6. HDAC6 and Chk1 Physically Interact

As previously discussed, MSH2 is currently the only reported target of HDAC6’s E3 ligase activity, whose protein levels are regulated by HDAC6 via sequential deacetylation and ubiquitination [22]. HDAC6 directly interacts with MSH2, so it stands to reason that Chk1 may directly interact with HDAC6 as well. Specifically, we hypothesize that HDAC6 directly interacts with and ubiquitinates Chk1, leading to degradation of Chk1. If this hypothesis is correct, enhanced Chk1 stability in HDAC6 knockdown and knockout cells would be due to the elimination of a critical E3 ligase for Chk1 degradation.

We began by assessing whether HDAC6 and Chk1 interact. As shown in Figure 9A,B, reciprocal co-immunoprecipitation assays indicate that overexpressed Flag-tagged Chk1 and HA-tagged HDAC6 interact in 293T cells. To investigate whether this HDAC6 and Chk1 interaction is relevant to normal cellular levels of these two proteins, we immunoprecipitated endogenous HDAC6 in 293T cells and found that endogenous Chk1 was able to co-immunoprecipitate with HDAC6 (Figure 9C). To ensure that this is a direct interaction and not the result of cofactor participation, bacterially purified GST-tagged HDAC6 and His-tagged Chk1 were utilized in an in vitro co-immunoprecipitation assay. As shown in Figure 9D, glutathione-agarose bound GST-HDAC6 was able to pull-down His-Chk1, suggesting that HDAC6 and Chk1 physically interact with each other.

Assured that HDAC6 and Chk1 can physically interact in both cellular and cell-free assays, we sought to determine which region of HDAC6 is responsible for the interaction with Chk1. From N-terminus to C-terminus, HDAC6 consists of an E3 ligase-containing DAC1 catalytic domain, the deacetylase-containing DAC2 catalytic domain, a cytoplasmic-anchoring SE14 motif, and a ubiquitin-interacting zinc finger motif (ZnF) [3]. The available Flag-tagged deletion mutants of HDAC6 [49] were shown as indicated in the Figure 10B scheme. These mutant proteins were overexpressed along with Myc-Chk1 in 293T cells. As shown in Figure 9A, the DAC1 domain (1–503) displayed the strongest interaction with Myc-Chk1, followed by full length HDAC6 (1–1215). There was minimal DAC2 domain binding (448–840), while the C-terminal mutant (840–1215) failed to interact with Myc-Chk1. The two deletion mutant constructs that displayed the strongest interaction with Myc-Chk1 shared a common feature; they contain the DAC1 E3 ligase domain (Figure 10).

In the reciprocal experiment, we utilized Chk1 deletion mutant constructs. From N-terminus to C-terminus, Chk1 consists of a kinase domain, an SQ motif containing phosphorylation sites Serine 317 and Serine 345, a CM1 motif (which serves as a nuclear export sequence), and a CM2 motif (which serves as a nuclear localization sequence) [50]. We used full length Chk1, along with an N-terminal fragment and a C-terminal fragment. After overexpressing these constructs along with Flag-HDAC6, we found that every construct was able to interact with Flag-HDAC6 (Figure 10C,D). These results may imply a plausible multi-faceted interaction between Chk1 and HDAC6, but the exact regions that participate in this interaction remain unidentified.

## 4. Discussion

Here, we describe a novel mechanism through which HDAC6 regulates Chk1 protein levels in NSCLC cells. Chk1 is a critical Ser/Thr kinase central to the maintenance of genomic integrity, activated in response to single- and double-strand DNA breaks by increasingly specific inhibitors are developed Chk1 activity may prove too broad for this treatment direction to achieve FDA approval. In light of these failures in the clinic, multiple groups have suggested an alternative method of targeting Chk1; swinging the pendulum in the opposite direction and constitutively activating Chk1 [23].

Spearheaded by Dr. Tony Hunter and Dr. Youwei Zhang, this alternative Chk1 targeting strategy is the result of major insights made in Chk1 spatio-temporal regulation. ATR activates Chk1 in the nucleus, specifically on the chromatin [23] and, to a lesser extent, ATM [51]. Active Chk1 phosphorylates numerous downstream targets to arrest the cell cycle in S/G2/M, allowing time for DDR proteins to repair the damage. Thus, destruction of Chk1 activity would impair the checkpoint and promote apoptosis, and clinical trials targeting Chk1 are ongoing. However, these trials have not proceeded past Phase II due to multiple factors, including a lack of antitumor efficacy, excessive toxicity, and a concomitant activation of compensatory ATM and ERK1/2 signaling [52]. Chk1 loss is embryonic lethal in mice [43], where Chk1 can arrest the cell cycle (through phosphorylation of CDC25A and CDC25C) and activate homologous recombination (HR) proteins Rad51 and BRCA2 [53,54]. However, phosphorylation by ATR on Serine 345 can also target Chk1 for nuclear export via Crm1 [55], inactivation by phosphatase PP2A [56], and ubiquitination by nuclear E3 ubiquitin ligase CDT2 [34]. The fraction of Chk1 that is exported to the cytosol is still active, but in this compartment it can only perpetuate cell cycle arrest through maintain phosphorylation of CDC25A and CDC25C [23]. Cytosolic Chk1 can then be ubiquitinated by E3 ubiquitin ligase Fbx6 [32], which allows for release from cell cycle arrest. While this model of Chk1 regulation has been widely accepted, it does not necessarily conflict with our discovery of a new Chk1 E3 ligase. Tumor suppressor p53 has over ten identified E3 ligases [37], and having multiple regulators fits the need for the cell to have tight and highly regulated control of DNA damage responders. Here, we are proposing that IR-induced DNA damage activates ATR as previously described, which activates canonical downstream target Chk1. This active Chk1 acts by phosphorylating canonical downstream target CDC25C, an inhibitory phosphorylation that renders CDC25C inactive and prevents it from allowing progression through the G2 checkpoint, thus initiating G2 arrest. In cells that are HDAC6 competent, HDAC6 will target Chk1 through its ubiquitin E3 ligase activity, marking Chk1 for degradation and terminating the G2 arrest. Conversely, loss of HDAC6 will render cells unable to resolve this DNA damage-induced G2 arrest.

Despite multiple groups acknowledging that constitutive Chk1 activation may be detrimental to cancer cell viability [38,39], our group is the first to propose a method for accomplishing this aim. In multiple cell lines, we have demonstrated that HDAC6 knockdown increases basal Chk1 levels (Figure 7), and that further treatment of these cells with DNA damaging agents exacerbates Chk1 activation and persistence in parallel to their inability to resolve breaks (Figure 3, Figure 4 and Figure 5). Upon recognition of Chk1, we hypothesize that HDAC6 ubiquitinates and targets Chk1 for degradation. HDAC6 protein levels are elevated in NSCLC tumors when compared to control tissues [11], and in our cell lines, loss of HDAC6 correlates with both an increase in total and active Chk1 (Figure 3, Figure 4 and Figure 5) as well as a radiosensitivity reliant upon Chk1 activity (Figure 6). Inhibition of HDAC6’s E3 ubiquitin ligase activity could potentially eliminate a mechanism NSCLC is reliant upon, resulting in cell death.

While E3 ligase inhibitors are commercially available as research tools, the majority are nonspecific due to the ubiquity of the pathways E3 ligases participate in, and thus there is virtually no interest in translating this strategy into the clinic. Unfortunately, we cannot circumvent this shortcoming with clinically relevant HDAC6-specific inhibitors, as both ACY-1215 and ACY-241 are designed to solely inhibit the deacetylase activity of the DAC2 domain. Consistently, we have shown that both inhibitors cannot increase the level of Chk1 upon IR (Appendix A), suggesting that the level of Chk1 may only be controlled by HDAC6’s E3 ligase activity, but not HDAC6’s deacetylase activity. Our research supports the design of an inhibitor able to target the DAC1 domain of HDAC6, or potentially both the DAC1 and DAC2 domain, to achieve total HDAC6 inhibition. Development of such an inhibitor, if truly HDAC6-specific and fitting the patient tolerability profile seen with current HDAC6 deacetylase inhibitors, would provide a strategy to sensitize NSCLC tumors to conventional DNA damaging agents. Alternatively, methods to target the upstream regulators of HDAC6 could indirectly decrease its protein levels and reduce deacetylase and E3 ubiquitin ligase activity. For instance, tamoxifen treatment of MCF-7 breast cancer cells prevented estradiol-stimulated HDAC6 accumulation and α-tubulin deacetylation [57], and nitric oxide (NO) is able to increase the acetylation of α-tubulin in A549 cells by preventing the essential S-nitrosylation of HDAC6 [58]. More directly, Cullin 3SPOP has been reported to destabilize HDAC6 via polyubiquitination, and this interaction has suggested that SPOP serves a tumor suppressor function through this mechanism [59]. Our recent studies have shown that USP10 is an HDAC6 stabilizer [11], and thus targeting USP10 would decrease the level of HDAC6. Further research is needed to identify the full spectrum of proteins that regulate HDAC6 stability, especially due to the relatively long half-life of HDAC6. From the above studies, we may be able to inhibit the upstream HDAC6 regulators to suppress HDAC6’s E3 ligase activity. Although we showed that HDAC6 inhibitors (ACY-241 and ACY-1215) were not able to regulate Chk1’s level upon IR (Appendix A), more work is warranted to decipher whether HDAC6’s deacetylase activity cooperates with its E3 ligase activity to modulate Chk1.

## Figures and Tables

**Figure 1 cells-09-02237-f001:**
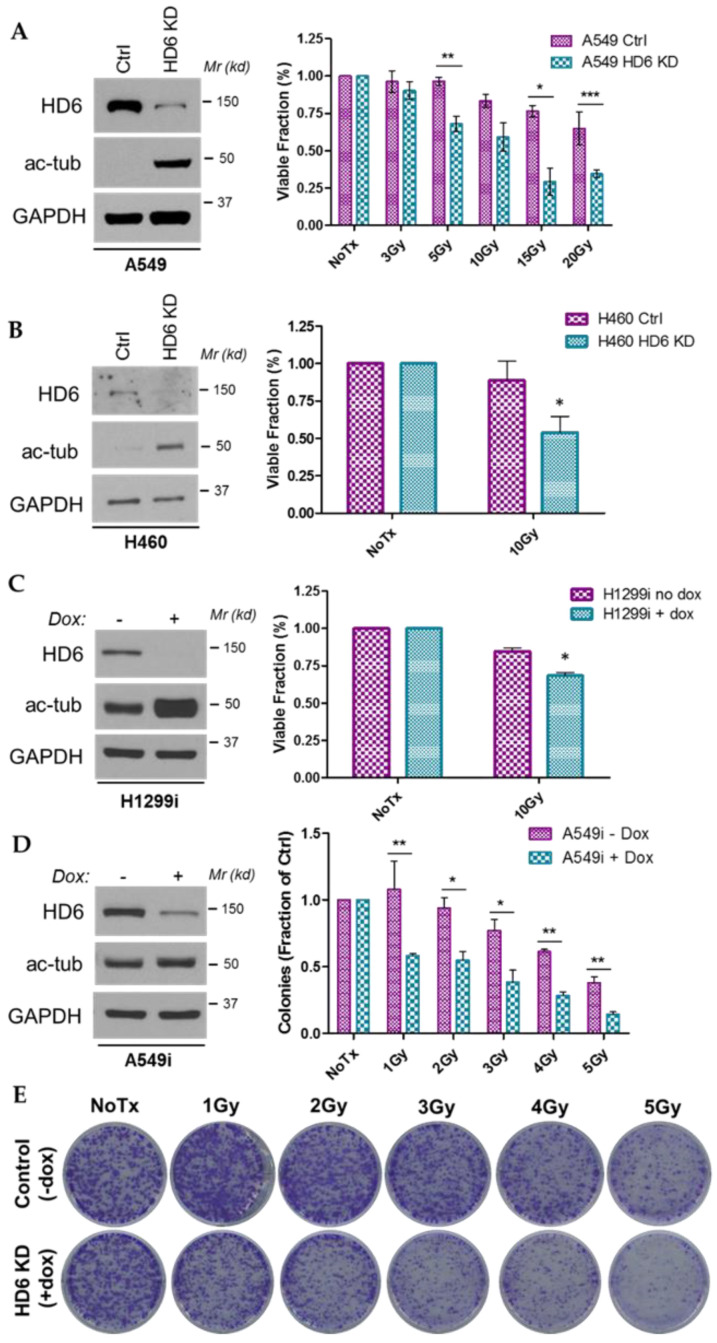
**HDAC6 knockdown (KD) sensitizes several NSCLC cell lines to ionizing radiation (IR).** (**A**) Smaller fractions of viable cells were found in the A549 HDAC6 KD (HD6 KD) cell line as compared to the A549 control cell line upon IR treatment. Left panel: Western blot confirming HDAC6 knockdown in A549 cells. Right panel: 120 h post-IR, A549 control and HDAC6 stable knockdown cells were suspended in trypan blue. The number of unstained cells (viable), stained cells (non-viable), and total numbers were recorded. Three biological replicates are graphed. Student’s *t*-tests were performed. * *p* = 0.0122, ** *p* = 0.0099, *** *p* = 0.0021. (**B**) Smaller fractions of viable cells were found in the H460 HD6 KD cell line as compared to the control cell line upon IR treatment. Left panel: Western blot confirming HDAC6 knockdown in H460 cells. Right panel: H460 stable HDAC6 knockdown cells were either left untreated, or treated with 10 Gy IR. 120 h later, trypan blue staining was conducted as described in (**A**). Student’s *t* tests were performed; * *p* = 0.0154. (**C**) Smaller fractions of viable cells were found in the H1299 HDAC6 inducible cell line (H1299i, Dox+) as compared to the control cell line (H1299i, Dox−). Left panel: Western blot confirming inducible HDAC6 knockdown in H1299i cells pre-treated with doxycycline (Dox) for two weeks. Right panel: H1299i cells were either left untreated, or treated with 10 Gy IR. 120 h later, trypan blue staining was conducted as described in (**A**). Student’s *t* tests were performed, * *p* = 0.0002. (**D**) Decreased numbers of colonies were observed in A549 HDAC6 inducible knockdown cells (A549i, Dox+) as compared to the control cells (A549i, Dox−). Cells were plated in 6-well plates at a concentration of 300 cells/well, incubated for 24 h, and irradiated with the indicated dose. 14 days later, cells were stained with crystal violet. Student’s *t* tests were peformed; * *p* < 0.02, ** *p* < 0.005. Error bars, S.D. (**E**) Representative images from the experiments performed in (**D**).

**Figure 2 cells-09-02237-f002:**
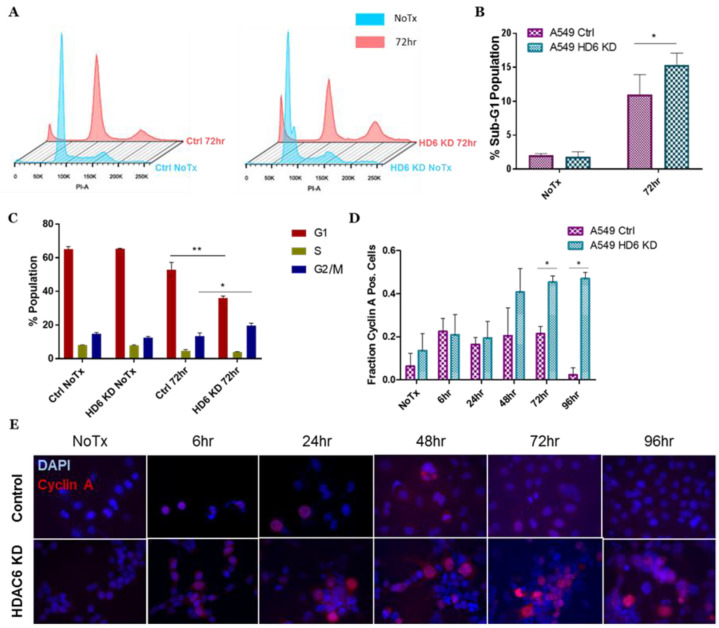
**HDAC6 knockdown A549 cells arrest at G2/M phase post-IR.** (**A**) A549 control cells (Ctrl) and A549 HDAC6 stable knockdown cells (HD6 KD) were either left untreated (No Tx, blue histograms) or irradiated with 10 Gy, incubated for 72 h (red histograms), harvested, ethanol fixed, and stained with PI. Cells were then analyzed via flow cytometry. (**B**) Analysis of the fractions of sub-G1 cells present in the experiments described in (**A**). Statistical significance was assessed using Student’s *t* test, with * *p* < 0.05. (**C**) Analysis of the cell cycle distribution from the experiments described in (**A**). (**D**) A549 control and HDAC6 stable knockdown cells were treated with 10 Gy IR at the indicated time points, and then the cells were stained with immunofluorescence for cyclin A. Results of cyclin A positivity from three biological replicates in these two cell lines were assessed for statistical significance using Student’s *t* test, with * *p* < 0.05 and ** *p* < 0.01. (**E**) Representative images of the data graphed in (**D**).

**Figure 3 cells-09-02237-f003:**
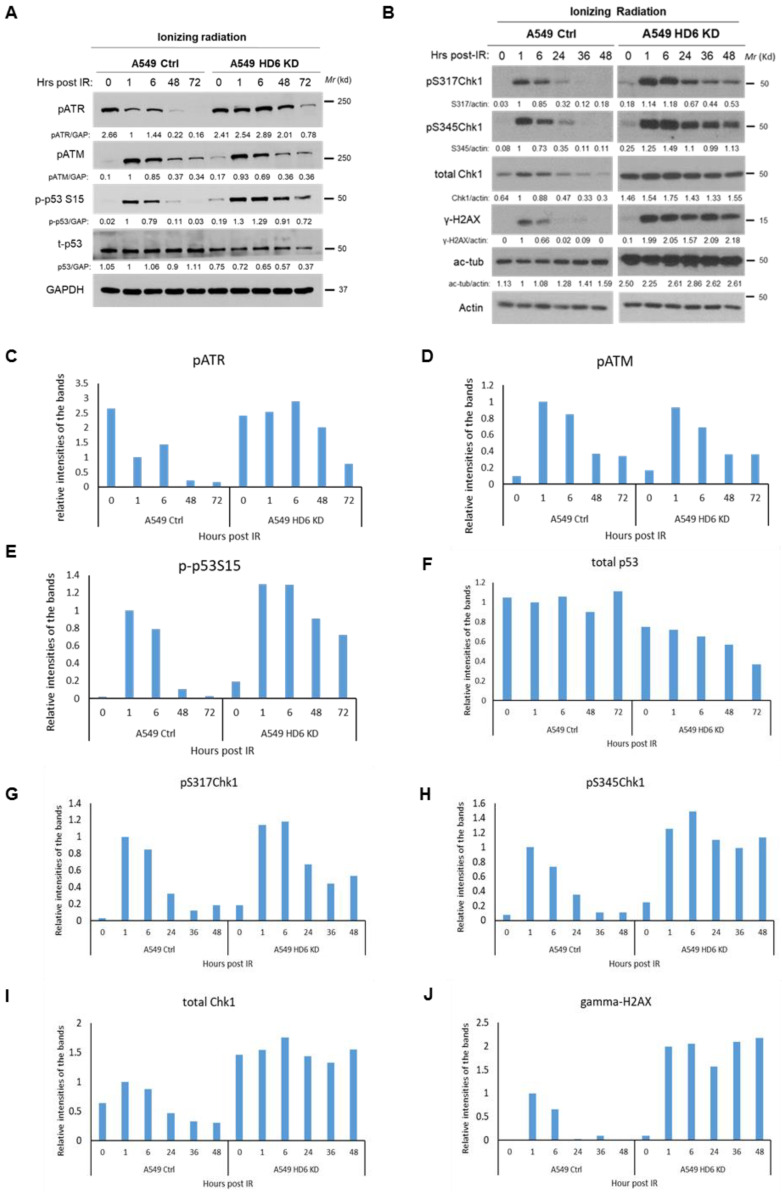
**Examination of DDR markers in A549 control and A549 HDAC6 knockdown cells post-IR.** A549 control and HDAC6 knockdown cells were irradiated with a dose of 10 Gy, harvested at the indicated time points, and the lysates were analyzed via Western blot by a series of antibodies: anti-pATR, anti-pATM, anti-p-p53S15, anti-total p53, and anti-GAPDH in (**A**) or by anti-pS317Chk1, anti-pS345Chk1, anti-total Chk1, anti-γ-H2AX, anti-acetylated tubulin (ac-tub) and anti-actin in (**B**). Blots were quantified via ImageJ, and reported quantification was normalized to the signal of the A549 control cells 1 h post-IR. The bar graphs for the expression of indicated DDR proteins are shown in (**C**–**J**).

**Figure 4 cells-09-02237-f004:**
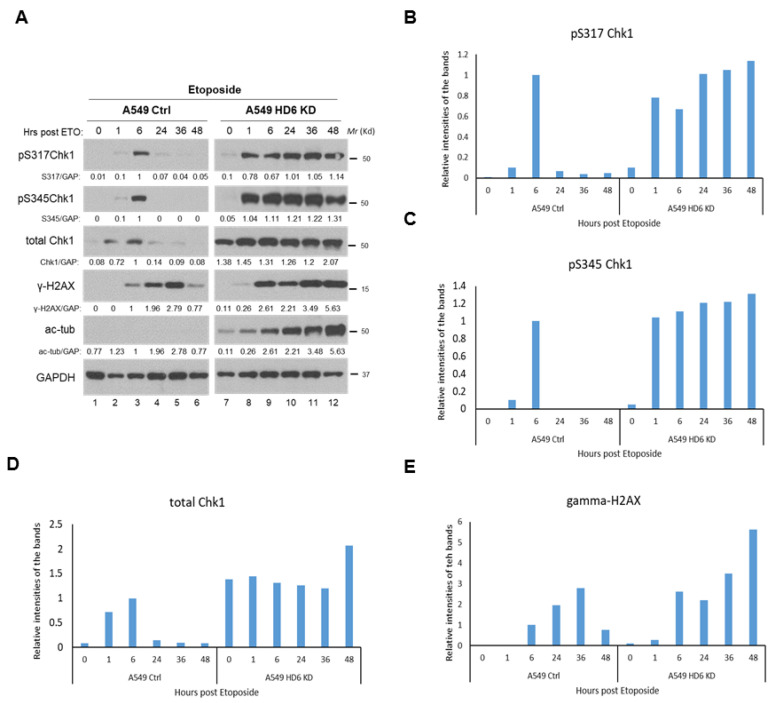
**Examination of DDR markers in A549 control and A549 HDAC6 knockdown cells post-Etoposide treatment.** (**A**) A549 control and HDAC6 knockdown cells were treated with 20 μM Etoposide, harvested at the indicated time points, and the lysates were analyzed via Western blot by a series of antibodies: anti-pS317Chk1, anti-pS345Chk1, anti-total Chk1, anti-γ-H2AX, anti-acetylated tubulin, and anti-GAPDH. Blots were quantified via ImageJ, and reported quantification was normalized to the signal of the A549 control cells 6 h post-Etoposide. The bar graphs for the expression of indicated DDR proteins are shown in (**B**–**E**).

**Figure 5 cells-09-02237-f005:**
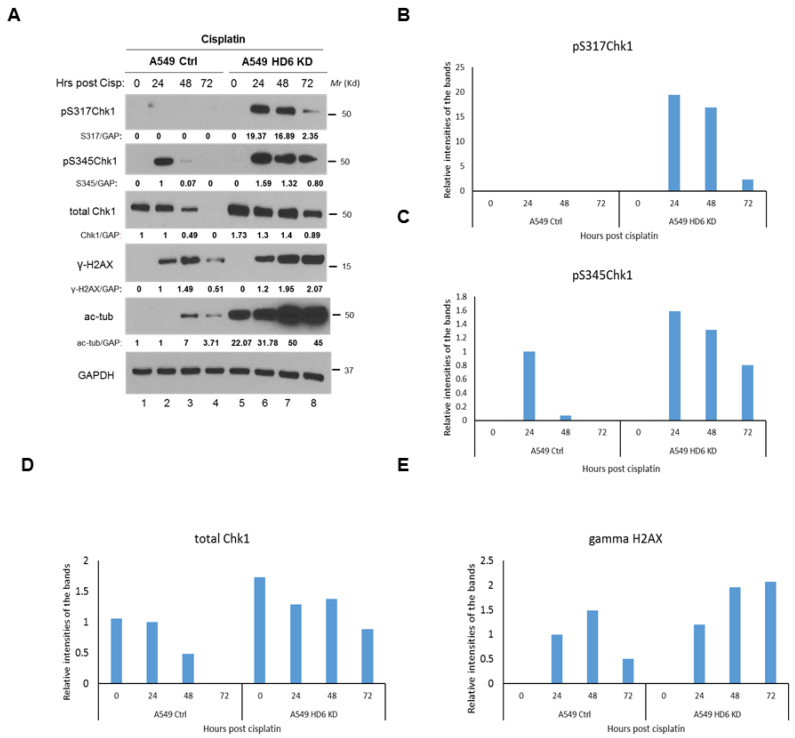
**Examination of DDR markers in A549 control and A549 HDAC6 knockdown cells post-Cisplatin treatment.** (**A**) A549 control and HDAC6 knockdown cells were treated with 10 μM Cisplatin, harvested at the indicated time points, and lysates were analyzed via Western blot by a series of antibodies: anti-pS317Chk1, anti-pS345Chk1, anti-total Chk1, anti-γ-H2AX, anti-acetylated tubulin, and anti-GAPDH. Blots were quantified via ImageJ, and reported quantification was normalized to the signal of the A549 control cells 24 h post-Cisplatin treatment, except for the pS317Chk1 bands whose normalization was chosen randomly. The bar graphs for the expression of indicated DDR proteins are shown in (**B**–**E**).

**Figure 6 cells-09-02237-f006:**
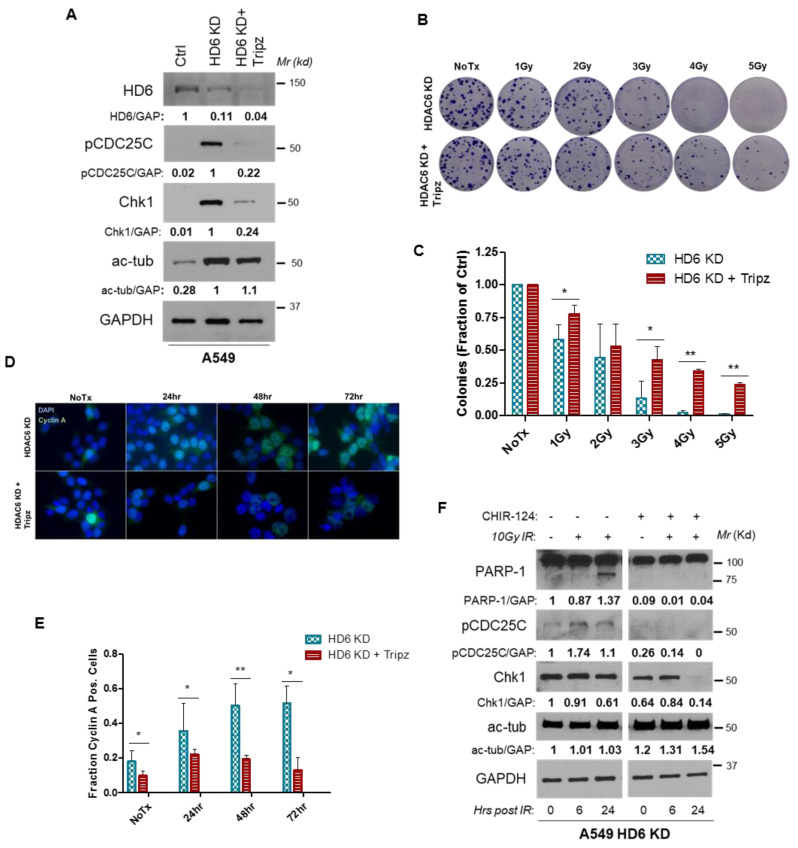
**The depletion or inhibition of Chk1 in HDAC6 knockdown A549 cells restores radio-resistance.** (**A**) Establishment of Chk1 knockdown cells in A549 HDAC6 knockdown cells (termed HDAC6KD+Tripz) was described in the Methods. Anti-HDAC6 and anti-Chk1 Western blotting analyses were performed to confirm the efficacy of HDAC6 and Chk1 double knockdown. Anti-pCDC25C, anti-ac-tub, and anti-GAPDH Western blotting analyses were also performed. Representative images of the data graphed Blots were quantified via ImageJ. For HDAC6, the reported quantification was normalized to the signal of the Control group. For the rest of the proteins, the reported quantification was normalized to the signal of the HDAC6 knockdown group. HDAC6KD+Tripz and HDAC6KD cells were plated in triplicate at a concentration of 150 cells/well and treated with the indicated dose of radiation. Cells were incubated for 12 days, fixed with crystal violet. The representative images are shown in (**B**). The colonies were quantified. Student *t* test, * *p* < 0.05, ** *p* < 0.0008. A bar graph presenting the above colony formation assays is shown in (**C**). HDAC6KD+Tripz and HDAC6KD cells were treated with 5Gy IR, and the immunofluorescence for Cyclin A was conducted. Representative images are shown in (**D**). Results of Cyclin A positivity from three experimental replicates, with significance assessed using student’s *t* test, with * *p* < 0.01, ** *p* = 0.0001. A bar graph representing cyclin A positive cells is shown in (**E**). (**F**) A549 HDAC6 stable knockdown cells were pre-treated with 0.25 μM of potent Chk1 inhibitor CHIR-124 prior to 10 Gy irradiation. At the indicated time points, cells were harvested and probed for the indicated proteins via Western blot. Blots were quantified via ImageJ, and normalized to the signal of the 0 h timepoint of the HDAC6 knockdowns treated with IR alone.

**Figure 7 cells-09-02237-f007:**
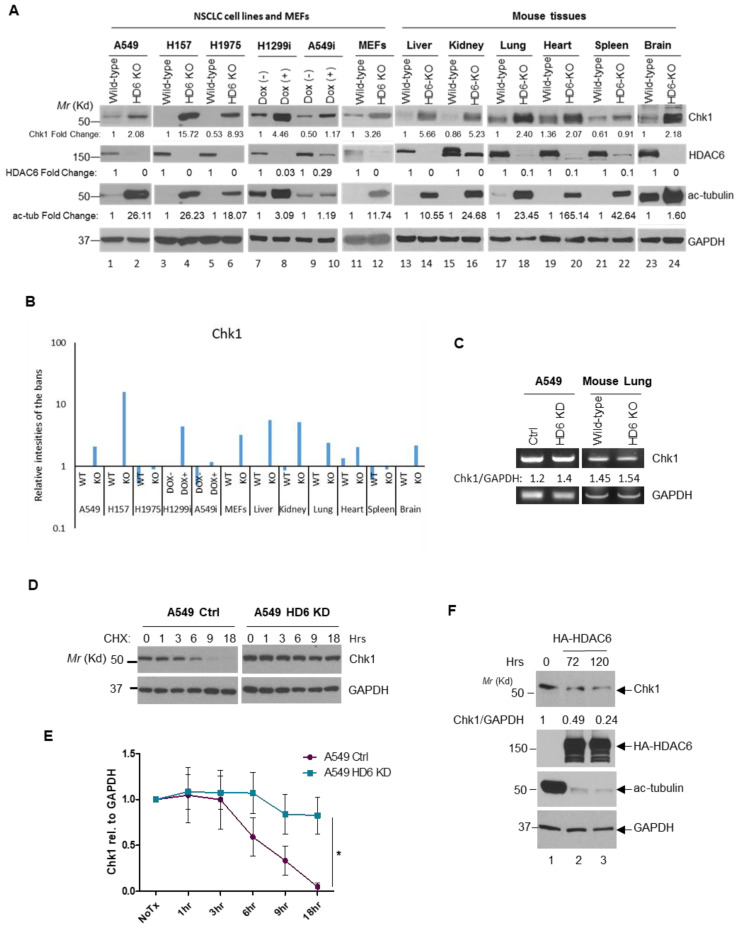
**HDAC6 influences Chk1 protein stability.** (**A**) (From left to right) A549 HDAC6 KO cells generated with the CRISPR-Cas9 system. H157 and H1975 HDAC6 KO cells generated with the CRISPR-Cas9 system. H1299 and A549 inducible HDAC6 knockdown cells (termed H1299i and A549i, respectively) pre-treated with doxycycline for two weeks. Mouse embryonic fibroblasts (MEFs) harvested from age-matched wild-type and HDAC6 KO mice (both from a C57Bl/6 background). Liver, kidney, lung, heart, spleen, and brain tissue harvested from age-matched wild type and transgenic HDAC6 KO mice (both from a C57Bl/6 background). All cell lines and tissues were lysed and analyzed via Western Blot for Chk1, HDAC6, acetylated tubulin, and GAPDH expression. The bands for Chk1, HDAC6, and ac-tubulin were quantified. (**B**) The relative expression of Chk1 from the cell lines and mouse tissues has been shown in a bar graph with logarithmic y-axis. (**C**) Chk1 RT-PCR was used to determine whether the depletion of HDAC6 influences Chk1 mRNA levels in A549 control and HDAC6 stable knockdown cells, as well as WT and HDAC6 knockout murine lung tissue. GAPDH RT-PCR was used as a loading control. (**D**) (Above) A549 stable knockdown cells were treated with 10 μg/mL cycloheximide (CHX), harvested at the indicated time points, and analyzed via Western blot. Representative Western blot of Chk1 and GAPDH from the trials was used to determine Chk1 half-life. (**E**) The average intensity of Chk1 relative to GAPDH expression from three independent experiments was obtained (via ImageJ) and graphed. The student *t* test was performed. Error bars represent S. D. * *p* < 0.05. (**F**) 293T HDAC6 knockout cells were plated, and 24 h later were untransfected or transfected with 2.4 μg HA-tagged HDAC6, harvested at the indicated time points and probed for the indicated proteins. HA-HDAC6 was detected using anti-HDAC6 antibody. Fold-change in Chk1 expression was evaluated via ImageJ.

**Figure 8 cells-09-02237-f008:**
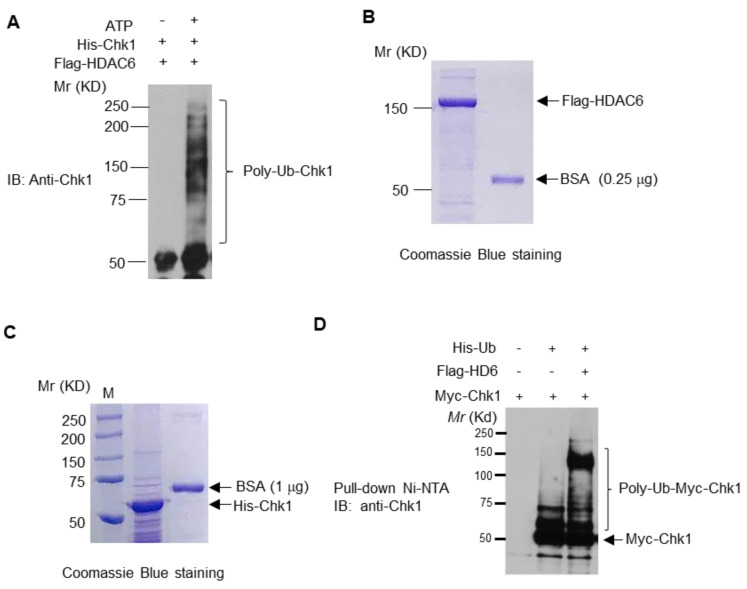
**HDAC6 ubiquitinates Chk1 in vitro and in vivo.** (**A**) HDAC6 ubiquitinates Chk1 in vitro. (**A**) The in vitro Ub assays were carried out in the presence of E1, E2, Ub, His-Chk1, Flag-HDAC6 in the absence or presence of ATP. The reactions were incubated at 37 °C for 2 h, denatured at 95 °C for 5 min, then added protein loading buffer. The reactions were loaded into SDS-PAGE followed by Western blotting analysis with the anti-Chk1 antibody. The detailed protocol is described in the Methods and Zhang et al. [22] (**B**) The Flag-HDAC6 was transfected into 293T cells. The Flag-HDAC6 protein was then isolated anti-Flag M2 agarose followed by Coomassie Blue staining. (**C**). His-Chk1 was purified from *E. coli* with Ni-NTA beads followed by Coomassie Blue staining. (**D**) HDAC6 ubiquitinates Chk1 in vivo. Mammalian expression vectors containing Myc-Chk1, Flag-HDAC6, and His-Ub were transfected into 293T cells. Cells were incubated for 48 h, harvested, and passed through a Ni-NTA column to pull down for His-Ub. Bound proteins were subsequently eluted from the columns, run on an SDS-PAGE gel, and probed for Chk1.

**Figure 9 cells-09-02237-f009:**
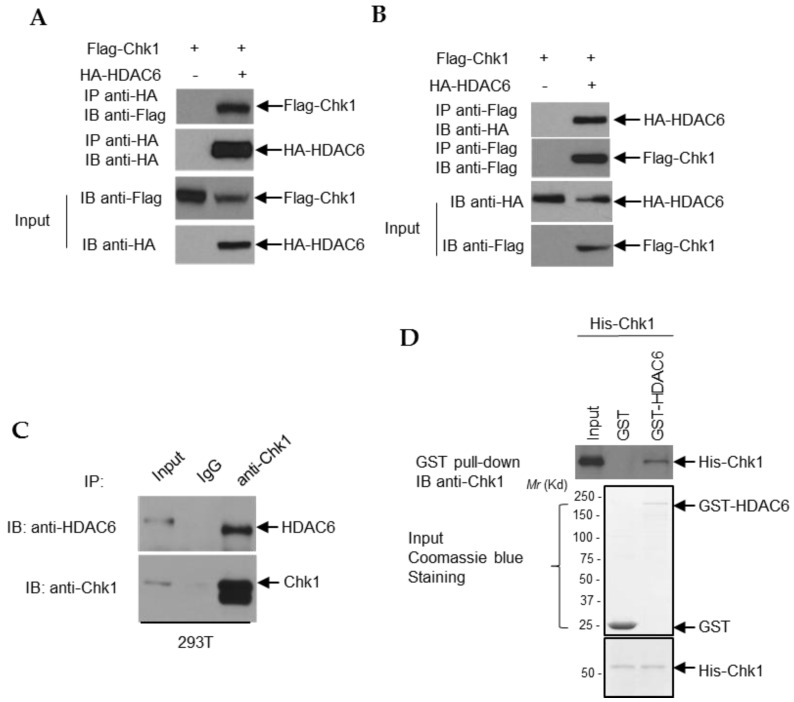
**HDAC6 and Chk1 physically interact.** (**A**,**B**) Mammalian expression vectors containing Flag-Chk1 and HA-HDAC6 were transfected into 293T cells with PEI. 48 h after overexpression, cells were harvested in lysis buffer, incubated with either HA-coated (**A**) or Flag-coated (**B**) agarose beads, and the resultant immunoprecipitated protein was run on an SDS-page gel and probed for the reciprocal tag. (**C**) 293T lysates were probed with anti-Chk1 antibody complexed with protein A/G beads, the beads were washed, and the resulting milieu probed for HDAC6 to detect an endogenous interaction between Chk1 and HDAC6. (**D**) His-Chk1 was overexpressed in *E. coli*. His-Chk1 was purified with Ni-NTA agarose beads. Then, GST and GST-HDAC6 were overexpressed in *E. coli*, and GST-tagged protein was pulled-down and purified by glutathione-agarose. Purified His-Chk1 was incubated with either glutathione agarose-bound GST or GST-HDAC6, and then bound proteins were eluted. The samples were subjected to SDS-PAGE and Western blot analysis.

**Figure 10 cells-09-02237-f010:**
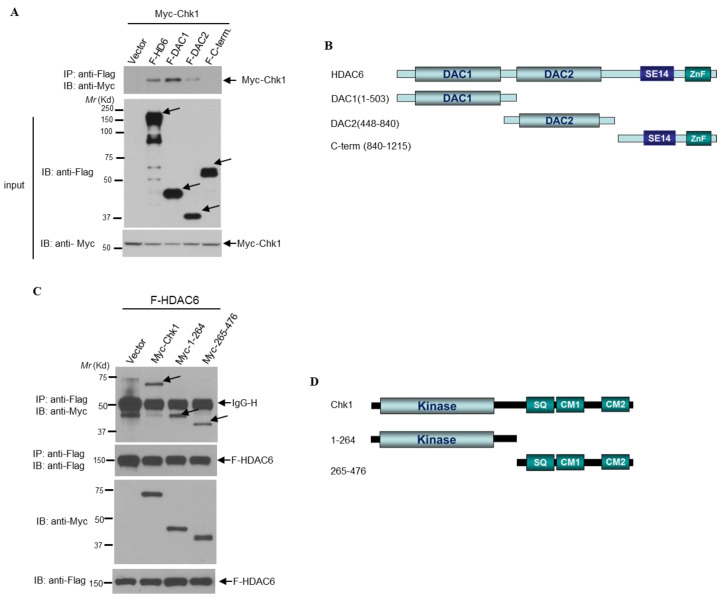
**HDAC6 interacts with Chk1 via its DAC1 domain.** (**A**) The indicated Flag-tagged HDAC6 deletion mutant constructs were transfected into 293T cells along with Myc-Chk1. 48 h later, cells were lysed, and lysates were pulled down for Flag. (**B**) Schematic of the Flag-tagged HDAC6 deletion mutant constructs used for the co-immunoprecipitation in (**A**). (**C**) The indicated Myc-tagged Chk1 deletion mutant constructs were transfected into 293T cells along with Flag-HDAC6. 48 h later, cells were lysed, and lysates pulled down for Flag. (**D**) Schematic of the Myc-tagged Chk1 deletion constructs used for the co-immunoprecipitation in (**C**).

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
