# Peer review of "HDAC6 Regulates Radiosensitivity of Non-Small Cell Lung Cancer by Promoting Degradation of Chk1"

_cells, 2020, doi:10.3390/cells9102237_

Round 1

Reviewer 1 Report

Authors describe a novel mechanism about HDAC6 regulates Chk1 protein levels in NSCLC cells. Many experiments were designed and performed. I think the manuscript is interesting to readers, however, it would be better if the full spectrum of proteins for regulating HDAC6 stability was identified in this study.

Reviewer 2 Report

This work is concerned with the molecular mechanisms underlying the sensitization of NSCLC cells to ionizing radiation via HDAC6 inhibition and stabilization of Chk1.

There are three important imperfections in this article:

  • The authors showed that NSCLC cells are more sensitive to the radiation due to the stabilization of Chk1 during genetic suppression of HDAC6. However, it remains unclear whether the effects shown are unique only with genetic suppression of HDAC6? Can similar results (decreased colony formation, increased apoptosis, accumulation of Chk1) be achieved using chemical inhibitors of HDAC6 (tubacin or CAY10603), as chemical inhibitors are more appropriate for the cancer treatment.
  • The major premise of this work, stated as “Chk1 is a HDAC6’s second potential E3 ligase substrate”, is not supported by the experimental approach. The paper provides a lot of convincing evidence of the HDAC6 and Chk1 interaction. The results reveal the physical interaction between the ubiquitinated Chk1 and HDAC6. However, the presence of ubiquitinated protein in complexes with HDAC6 does not prove the role of HDAC6 as an E3 ligase. HDAC6 is an ubiquitin-binding protein which binds to and facilitates the transport of polyubiquitinated misfolded proteins along microtubules to aggresome. To prove the activity of HDAC6 as E3 ligase, it is required to perform the Chk1 ubiquitination in vitro.

  • The authors state that enhanced Chk1 activity leads to preferential G2 arrest in HDAC6 knockdown. This result is not supported by a sufficient amount of data. To confirm the Chk1 inhibition effect on a decrease in G2 arrest,  authors should perform FACS analysis of HDAC6-KO cells using a Chk1 inhibitor.

The results reported in the manuscript are of high quality and represented by the appropriate number of replicates. Overall this is a solid contribution to the understanding of how HDAC6 inhibition can sensitize NSCLC cells to IR. Some issues need to be clarified and the addition of some further data, if available, would strengthen the paper.

Minor points:

  • Line 120: “... Independent clones were then cosen by induction with…” - “cosen” change to “chosen”
  • The section 2. Materials and Methods notifies that the viability was assessed as:% viable cells = [1.00 – (number of blue cells (non-viable) – number of total cells)] x 100. This formula assumes that 100% viability means a situation where the number of blue cells (non-viable) = number of total cells (In other words, 100% viability = all cells are not viable). There is a mistake in the formula. Moreover, in Figure 1 A-C it is not clear what is the unit of the y-axis (Fold or %). What is taken as the baseline?
  • The influence of HDAC6 inhibition for 72 hours on the NSCLC cells viability and the apoptosis level should be demonstrated. This control is valuable since the HDAC6 inhibition affects the tumor cells viability itself. Accordingly, it is necessary to separate the contribution of HDAC6-KO from the IR/HDAC6-KO contribution. Widely accepted fact is that inhibition of HDAC6 reduces the proliferation of tumor cells and induces apoptosis. Particularly, there are data of the induction of apoptosis in lung tumor cells under HDAC6 inhibition (Wang et al., 2016, 10.3892/or.2016.4811; Deskin et al., 2020, doi.org/10.1016/j.tranon.2019.11.001). However, there is some data about normal proliferation of A549 cells lacking functional HDAC6 (doi.org/10.1016/j.bbrc.2008.06.092).
  • Paragraph 2.4 describes the generation of H1299 inducible HDAC6 knockdown cells. However, paragraph 2.6 describes the application of A549 inducible HDAC6 knockdown cells. If the generation of A549 inducible HDAC6 knockdown cells was different? The appropriate method should be described. If the knockdown cell generation method was uniform for all cells, paragraph 2.4 should be corrected accordingly.
  • Whether doxycycline was added in the growth medium for 13 days  after irradiation?
  • The experimental conditions should be specified for FACS-histograms of blue color in the legend to Figure 2 A. 
  • LIne 571: “... 48 hours post-transfection, cells were treated with 150μg/mL and…” - Agent name is missing.
  • It is not indicated which antibodies against pATR, pATM, p53, and γ-H2AX were used.
  • Please specify the antibodies used for the WB of Figure 5D (anti-HDAC6 or anti-HA).
  • Legend to Figure 4C states that Figure 4C shows representative data averaged in Figure 4D. However, in Figure 4D, there is data concerning Cyclin A expression, while Figure 4C shows the gamma-H2AX fluorescence.

Reviewer 3 Report

The authors firstly demonstrated the role of HDAC6 in degradation of Chk1 through the ubiquitin-proteasome pathway in NSCLC. In this study, to investigate the role of HDAC6, the authors established both stable and inducible HDAC6 knockdown cells. In addition, they genetically and chemically inhibited Chk1 to evaluate the role of chk1 that was up-regulated by HDAC6 KD in radio sensitivity. The authors also demonstrated that the DAC1 domain plays predominant role in the interaction between HDAC6 and Chk1 in HDAC6 KO cells established by using genome editing. This study is well designed and the finding will be interesting to the reader of this journal. Only a few concerns may be addressed before publication.

  1. PI-based cell cycle analysis is difficult to distinguish G2 and M phase. If the authors needs to discriminate between G2 and M phase, cells with phosho-H2AX at Ser10 should be detected as M phase in flowcytometry. If not, the authors may change "G2" to "G2/M".
  2. In supplementary figure, Phospho-p53 at Ser15 increased but total p53 levels did not changed. I wonder why p53 degradation was not inhibited since p53 phosphorylation at Ser15 inhibit the interaction between p53 and MDM2
  3. There seems to be no correlation between Fig.2D and Fig.2E because cyclin A positive cells is almost absent in control at 6-72 h. The authors may change typical image if possible.  In addition, I would like ask authors why they use cyclin A. Is Cyclin B not appropriate?

Reviewer 4 Report

In the manuscript named "HDAC6 Regulates Radiosensitivity of Non-Small Cell Lung Cancer by Promoting Degradation of Chk1," of the authors' Moses N et al. are summarising a novel mechanism of Chk1 regulation at the protein level and suggesting a possible treatment strategy for non-small cell lung cancer (NSCLC).

After reading the manuscript, I have some major as well as minor issues with this paper.

Major:

1) Chapter 2. Material and Methods have to be written in more detail; in the presented form, it is not sufficient. As well as described cells concentrations and time points should be unified across the whole manuscript (e.g. Material and Methods vs Results). 

2) Chapter 3.1 HDAC6 depletion sensitizes NSCLC to IR, in Figure 1A the authors provided viability graph for A549 stable HDAC6 knockdown cells, for different doses (from 3 to 20 Gy) at the same time (120 hrs/ 5 days) and stated "We found a significant and dose-dependent reduction in viability in the HDAC6 knockdown cells, and confirmed this reduction in H460 stable HDAC6 knockdown cells and H1299 inducible HDAC6 knockdown cells 120 hours post-10Gy radiation [Figure 1B, C]"

According to the Figure 1A graph, even 5 Gy dose would be efficient enough, and it is confusing for the readers how and why the authors chose for initial characterization of dose-dependency the only time of 120 hrs and not shorter time points. Could authors comment on these facts? 

I strongly recommend the authors to provide data for doses 3, 5, 10, 15 and 20 Gy for time points 6 hr, 24 hr, 48 hr, 72 hr and 96 hr for all three cell lines (A549 stable HDAC6 knockdown, H460 stable HDAC6 knockdown cells and H1299 inducible HDAC6 knockdown cells). It will be consistent with the rest of the results. 

3) Results presented in Figure 2 have no connection to previously described data. Authors without apparent justification used for the characterization of the cells cycle, by FACS, the cells irradiated 10 Gy and time 72 hr after irradiation. Then, in the same Figure, they again changed time points to 6 hr, 24 hr, 48 hr, 72 hr and 96 hr for immunofluorescence staining of Cyclin A in irradiated cells. Could the authors explain why?

4) In the Colony formation assay, authors ruled out the using of A549 and H460 stable knockdown cells and used A549 inducible HDAC6 knockdown cells instead. Could be the discrepancy between cell lines (suitable vs not suitable for assay) based on how knockdown cells were produced?

5) In the western blot analysis in Figure 3, the time points for harvesting the cells changed again. The authors used additionally 1 hour and 36 hours intervals after irradiation and did not included data from 72 hours and 96 hours after irradiation as well, but only in Figure 3A and 3B. In Figure 3C are missing time points: 1 hr, 6 hr and 36 hr. Authors have to unify the western blot data in Figure 3 and provide the results for intervals 1 hr, 6 hr, 24 hr, 36 hr, 48 hr, 72 hr and 96 hr. Also please, provide the quantification of western blots in Figure 3A-C.

6) By comparing data sets from Figure 3, it appears that γH2AX is present in the A549 HD6 KD cells for a very long period, even 72 hours after irradiation and also in the cells after irradiation combined with treatment (etoposide or cis-platine). Could authors comment on that in the context of DNA damage response (DDR) and HDAC6 knockdown?

7) In Chapter 3.4 authors compare the HDAC6 KD + Tripz cells to the single HDAC6 KD cells by performing immunofluorescence staining with Cyclin A. Why authors used 5 Gy (written in the description of Figure 4) of γ-irradiation instead of 10 Gy as they were using in the rest of the experiments?

Also, could authors provide at least the immunofluorescence images for time point 6 hr and 96 hr, the same as in Figure 2? As for western blot part of Figure 4 (E-F), please provide quantifications.

8) The authors stated (row 312): "...were unable to assess the full cell cycle distribution of our HDAC6 KD + 312 Tripz cells using PI staining..." not able to use FACS for distinguished between the phases of the cell cycle. Did authors consider to use pKi67 antibody for immunofluorescence instead of FACS analysis? Could the authors provide comment on that?

9) The authors stated (row 386): "...Chk1 protein levels were reduced by half and three-quarters, respectively, as compared to the control [Figure 5D]..." Please, provide the data from quantification, not an estimated guess.

Minor:

10) Using the name of the cell line HEK-293T vs 293T have to be unified.

11) Ubiquitylation Assay (Ub assay) is not mentioned in chapter Material and Methods.

12) In Figure 1 authors used for Colony formation assay 350 cells/ well and 14 days of cultivation, in Figure 4 150 cells/ well and 12 days of cultivation and finally, in Material and Methods they stated 300 cells/ well and 13 days incubation. Could the authors clarify or comment on these differences?

13) In Figure 3A, it is hard to distinguish each GAPDH bands (7 to 12), could authors exchange the exposition for a better one?

14) Figure 5A (3-7) provide a better western blot image for ac-tubulin as well as data quantification.

15) In the western blot analysis in Supplementary Figure 1, the authors omitted the time points: 24 hours, 36 hours and 96 hours after irradiation. Could authors unify the western blot data and provide the results for intervals 1 hr, 6 hr, 24 hr, 36 hr, 48 hr, 72 hr and 96 hr? Furthermore, provide the quantification of western blot.

15) In the western blot analysis in Supplementary Figure 2, the authors used 3 hours time point instead of 1 hour and again omitted the time points: 36 hr, 48 hr, 72 hr and 96 hr. Unify the western blot data as well and please, provide the quantification of western blot.

Row 480: the last sentence in Figure 7 description has a larger font.

Overall, I can not recommend the manuscript for publishing in the present for; major revision has to be done.

Round 2

Reviewer 2 Report

  • The authors have performed a number of important experiments that can significantly improve the presented work. However, the authors did not include the obtained experimental data (The in vitro Ub assays of Chk1) in the main text of the article, doubting the reliability of the data. Whilst those experiments support the results of an article, it seems to be an integral part of the research to confirm the main results of the paper, so they better be part of the original paper, and not a separate attachment only available to the reviewers. So suggestion is to augment the paper with the results obtained from the additional experiments, to make the paper claims more sound. 

  • The data on the inability of HDAC6 chemical inhibitors to prevent the IR-induced degradation of Chk1 suggest that some capabilities of HDAC6, other than to deacetylate the proteins (possibly E3-ligase activity), is involved in the degradation of Chk1. It makes sense to include the data and discuss this topic in the appropriate section.

  • The authors claim that they added to Figure 3C-D the data concerning the influence of HDAC6 inhibition on the NSCLC cells viability and the apoptosis level. However, there is no 3D figure in the article.

  • The legend to Figure 4 is inconvenient to read. The description of the double knockout cells generation should be placed at the beginning of the entire Figure legend, not only in paragraph (E)

Author Response

Please see our response to the reviewer in the uploaded file. Thanks.

Reviewer 4 Report

The manuscript named "HDAC6 Regulates Radiosensitivity of Non-Small Cell Lung Cancer by Promoting Degradation of Chk1," of the authors' Moses N et al. underwent some changes, but not in such extensive way, as it was requested.
First of all, the authors did not implement the responses to major points, of the reviewer's criticisms, in the main text of the manuscript, to improve it and made it more understandable and clear, not to mention additional citations. Instead, they only addressed the technical and methodological issues.
Also, some of the answers were more confusing than convincing.

In point 2) of Response to Reviewers, authors rationalized why they use selected time point (120 hours post-10Gy radiation). They stated that cells are viable and growing even 48 h after irradiation with 10 Gy dose and dying only after "repeated rounds of replication until the damage is too severe, and cell death occurs". Could it be that 120 h cells are dying naturally due to high concentration/ confluency?
Why is the 3 Gy treatment group mentioned in the last sentence of the answer?

By comparing additional Figure 1A (A549 stable KD MTT, 24 h, 10 Gy) and 1B ((A549 stable KD MTT, 24 h, 10 Gy), there is a discrepancy in relative viability. In Figure 1B, there is a significant decrease in relative viability, but not in Figure 1A (same dosage, same time point).
Is there a new cell line introduce H292 in the additional Figure 1E?

Point 3) the rationale behind using 72 h time point for FACS have to be included in the main text of the manuscript.

In point 5) of Response to Reviewers, the authors stated: "To maintain the consistency of Figure 3, we have moved our Cisplatin data (formerly Figure 3C) to the supplementary data section," but they did not. Instead, they only reorganized Figure 3. In Figure 3A, the data were quantified and normalized to GAPDH, but actin is presented.
Also, the rationale behind the structure of timecourses, from point 5) of Response to Reviewers, have to be included in the main text.

In general, the quantifications of the Western blot were provided, as reviewer requested, but only as hardly readable numbers inside Figures.
In Figure 4F, right part of the western blot needs to be re-quantified.

Row 96: the catalogue number of ChK1 antibody is mentioned twice.

Overall, despite some improvements, I can not recommend the manuscript for publishing in the present form; major revision has to be done.

Author Response

(The authors gave the same response as above.)

Round 3

Reviewer 2 Report

I have no additional comments

Author Response

Thank you. 

Reviewer 4 Report

The manuscript named "HDAC6 Regulates Radiosensitivity of Non-Small Cell Lung Cancer by Promoting Degradation of Chk1," of the authors' Moses N et al. has been improved after revisions.

The authors implemented reviewer criticism into a certain extent.

However, some minor points need to be addressed:

  • Figures 3-5 contain bar graphs of western blot quantification, but missing the error bars (assuming that the authors are presenting the average values of multiple biological repetitions and technical repeats). Also, the y-axis scales should be unified to make quantifications as comparable as possible.
  • in Figure 7 and the main text, mouse tissues (organ samples) are used, but in the Material and methods part, there is no mention about protocol or handling of experimental animals and tissues.  
  • in Appendix A, there are two figures labelled as Figure S2, first should be relabelled to Figure S
